# Landscape Enhancements in Apple Orchards: Higher Bumble Bee Queen Species Richness, but No Effect on Apple Quality

**DOI:** 10.3390/insects12050421

**Published:** 2021-05-08

**Authors:** Amélie Gervais, Marc Bélisle, Marc J. Mazerolle, Valérie Fournier

**Affiliations:** 1Département de Phytologie, Centre de Recherche et D’innovation sur les Végétaux, Université Laval, Québec, QC G1V 0A6, Canada; valerie.fournier@fsaa.ulaval.ca; 2Quebec Centre for Biodiversity Science, McGill University, Montreal, QC H3A 1B1, Canada; 3Centre SÈVE, Université de Sherbrooke, Sherbrooke, QC J1K 2R1, Canada; Marc.M.Belisle@USherbrooke.ca; 4Département de Biologie, Centre d’Étude de la Forêt, Université de Sherbrooke, Sherbrooke, QC J1K 2R1, Canada; 5Département des Sciences du Bois et de la Forêt, Centre D’Étude de la Forêt, Université Laval, Québec, QC G1V 0A6, Canada; marc.mazerolle@sbf.ulaval.ca

**Keywords:** Bombus, Apidae, bloom, hierarchical model, community composition, occupancy, pesticide, capture–mark–recapture (CMR)

## Abstract

**Simple Summary:**

Pollinators are essential to produce fruits in apple production. Bumble bees are among the most effective pollinators in orchards during the blooming season, yet they are often threatened by the high levels of pesticide use in apple production. Hedgerows and flower strips are infrequently sprayed by pesticides and are thus potentially good shelter for bumble bees. This study evaluated the influence of landscaping in the form of hedgerows and flower strips on the abundance and number of bumble bee species found in apple orchards. The number of bumble bee species found in orchards with hedgerows or flower strips was higher than in orchards without such landscape enhancements. Similarly, three species were more abundant in orchards with landscaping than orchards without those enhancements. Our work provides additional evidence that landscaping in the form of hedgerows and/or flower strips improves bumble bee presence in apple orchards and should therefore be considered as a means to enhance and ensure pollination within farms.

**Abstract:**

Bumble bees are among the most effective pollinators in orchards during the blooming period, yet they are often threatened by the high levels of pesticide use in apple production. This study aimed to evaluate the influence of landscape enhancements (e.g., hedgerows, flower strips) on bumble bee queens in apple orchards. Bumble bee queens from 12 orchards in southern Québec (Canada) were marked, released, and recaptured in the springs and falls of 2017 to 2019. Half of the 12 orchards had landscape enhancements. Apples were harvested in 2018 and 2019 to compare their quality (weight, diameter, sugar level, and seed number) in sites with and without landscape enhancements. Species richness, as well as the occurrence of three species out of eight, was higher in orchards with landscape enhancements than in orchards without such structures. The occurrence of *Bombus ternarius* was lower in orchards with high levels of pesticide use. Apples had fewer seeds when collected in orchards with landscape enhancements and were heavier in orchards that used more pesticides. Our work provides additional evidence that landscape enhancements improve bumble bee presence in apple orchards and should therefore be considered as a means to enhance pollination within farms.

## 1. Introduction

Apples are one of the most widely eaten fruits on the planet [1]. The adage “An apple a day keeps the doctor away” may have contributed to the fruit’s popularity, but in terms of health benefits, there is some truth in the saying. Apples are well known for their nutritional properties, such as high antioxidant levels, antiproliferative activity, inhibition of lipid oxidation, and cholesterol-lowering effects [2,3]. Global apple production totaled around 85 million tons in 2018, surpassed only by bananas and watermelons [1]. While China and the United States are, respectively, the two biggest apple producers, Canada lies in the top 35 apple-producing countries worldwide [1]. In Canada, the volume of apples produced in 2018 was by far greater than that of all other fruit crops [4]. 

One particularity of apple production is the cross-pollination required for fruit set. Not only must pollen grains come from another flower but they also usually need to come from another apple variety [5]. In fact, most cultivars are self-incompatible and thus require insects as pollen vectors for fruit production, or improved fruit quality [6,7]. Apples are usually larger, heavier, sweeter, and more symmetrical when pollinated by insects [8]. Deformed apples are linked to a lower number of seeds, generally caused by the absence of pollination [8,9]. Honey bees are usually used in orchards to ensure adequate pollination. For example, in the province of Quebec, apple producers rented 2300 hives for a total cost of CDN 152.9K in 2019 [10]. However, honey bees may not be the most efficient pollinators in apple orchards. In fact, early-spring apple blooming often limits pollination, especially by honey bees [11,12]. Temperatures rarely rise above 13 °C during the blooming period for ranges of latitudes in Quebec, which greatly reduces the diversity, abundance, and activity of flying insect pollinators [5]. Native bee species present in early spring are generally better adapted for ensuring apple pollination than honey bees, but the two groups can also work synergistically to increase fruit set [13]. 

Bumble bees (Hymenoptera: Apidae) are one of the most common native pollinators found in apple orchards in early spring, particularly during blooming. In fact, the queens of the genus *Bombus* are often the most efficient pollinators for apples, in terms of flower visitation rate, flower constancy, and pollen deposition [11,14,15,16]. In late summer and early fall, queens newly produced by colonies will mate with males and then hibernate as adults [17]. In spring, the newly emerged solitary queen requires a nesting location to establish her own colony and floral resources to feed herself and her future workers [17]. Apple orchards usually provide both of these resources during the blooming period. Yet, if queens are unable to secure sufficient food from around their nest, it can lead to the collapse of their colonies [17]. This issue is likely critical for queens nesting in or near apple orchards, where food resources substantially decline after apple bloom [17]. Although spring conditions determine colony survival through summer, fall conditions are equally critical as new queens need to ingest enough food to survive winter [17]. Within monocultures, such as those typically found in intensively managed orchards, food resources can be sparse during the fall and may thus jeopardize the survival of new queens [17]. 

In order to produce more marketable fruit, apple production often relies on the liberal use of pesticides to control pest insects and diseases [18]. However, pesticides are known to harm non-targeted, pollinating insects [19,20], such as bumble bee queens present in orchards [21]. In fact, apple production in the US scored fifth on the Environmental Working Group’s (EWG) “dirty list” of crops using the most pesticides, with only strawberries, spinach, kale, and nectarines considered more toxic [18]. In 2019, 17 pesticides, including five neonicotinoids and four fungicides, were detected in 379 apple juice samples evaluated by the Pesticide Data Program (PDP) [22]. Such combinations of pesticides are well known to be detrimental to bees [21,23,24,25,26]. For example, the fungicide imazalil, when used in mixture with common insecticides (namely: fipronil, thiamethoxam and cypermethrin), synergized the insecticidal toxicity on bumble bees [21]. Insecticides alone negatively affect bumble bee colony initiation [27], development [27,28,29,30,31], reproductive success [28,29,30,31], foraging ability, and homing success [32,33]. Unfortunately, apple growers are often constrained to juggle between lowering pesticide inputs and achieving a profitable yield. 

For growers, landscape enhancements may represent a way to increase the abundance of bumble bee queens in their orchards while simultaneously limiting the negative impact of pesticides on these insects. Often included in European agri-environmental schemes (AES), they can be as diverse as flower strips, riparian strips, flowered field margins, or hedgerows [34,35]. For pollinators, such landscape enhancements can provide food resources, nesting opportunities, and possible shelter from pesticide exposure, particularly for bumble bees [34,36,37,38,39,40,41,42,43,44]. For example, hedgerows around apple orchards in Spain were found to offer a rich succession of floral resources for bees, including bumble bees [39]. Moreover, in the US, buffer strips showing higher flower diversity were found to harbor or be visited by more beneficial insects, such as bumble bees [44].

In Quebec, Canada, apple orchards are mostly found in the southern portion of the province within landscapes often characterized by intensively managed row crops. Enhancements within such landscapes, and in or around apple orchards in particular, could thus be beneficial for bumble bees, and, ultimately, for growers. Several studies have investigated native bee communities in apple orchard agroecosystems [34,39,41]. Our study aimed to assess the relationships between landscape enhancements, bumble bee queen presence in apple orchards, and the quality of apples produced therein. More specifically, this study evaluated the effect of landscape enhancements on (1) bumble bee queen site occupancy; (2) bumble bee queen species richness; and (3) apple diameter, weight, sugar level, and seed number as a proxy of quality. We hypothesized that landscape enhancements benefit both bumble bees and apple quality. As pesticide use is currently a crucial component of apple orchard management, we also hypothesized that high intensity of pesticide use has a negative impact on bumble bee community composition and richness, yet a positive effect on apple quality.

## 2. Materials and Methods

### 2.1. Study Area

We measured the effects of landscape enhancements using 12 apple orchards located in two regions of southern Québec: Montérégie-Est and Estrie (Figure 1; [45]). Montérégie-Est is characterized by large-scale, intensive row cropping devoted to the production of maize, soybean, wheat, and other cereals [46], with small interspersed forest patches. Surface waters of this region are often contaminated by a vast array of pesticides, including neonicotinoids, many of them at concentrations deemed harmful to aquatic life [47,48]. In contrast, Estrie is mostly composed of small-scale dairy farms (hayfields and pastures) surrounded by substantial forest cover [49]. Each of the above two regions comprised six apple orchards, including three with existing landscape enhancements and three without. The nearest neighboring orchard was, on average, 19.2 ± 18.4 km (±SD) away, with the closest neighbors 2.8 km away (orchards #1 and #2), and the farthest 69.8 km away (orchards #8 and #7; Figure 1).

### 2.2. Bumble Bee Capture–Mark–Recapture

We opted for a capture–mark–recapture (CMR) approach to estimate site occupancy and species richness of bumble bee queens as this allowed us to alleviate biases related to imperfect detection probability (see below). We surveyed bumble bee queens during apple bloom in spring (May) and prior to hibernation in the fall (September–October), starting in spring 2017 and ending in spring 2019 (three springs, two falls). While bumble bees were captured only in the vicinity of apple trees during apple bloom, captures occurred throughout the orchard in the fall. During each season, orchards were visited three times. Each visit lasted for one hour and consisted of two different teams composed of two observers, for a total sampling effort of 12 h per orchard (3 visits/orchard × 2 teams/visit × 2 observers/team × 1 h/observer). At the beginning of each visit, the time of day and ambient temperature were noted. Each time a queen was caught, the timer was stopped during the marking process, resulting in some visits lasting more than 3 h (searching + handling). Usually, all visits within one orchard were performed on the same day, but on some occasions, mostly during the fall, consecutive visits could be separated by as much as two weeks (orchards #8 and 11; once). Bumble bee queens were captured by hand netting and then marked using a honey bee queen marking cage (Figure 2) and non-toxic, waterproof Craftsmart^®^ paint pens. Using 8 different colors, each individual queen was assigned a unique 3-color combination. We further recorded thorax width with a ruler (i.e., intertegular width to help with species identification), and abdomen color pattern before individuals were released.

Typically, the species found in the study area are *Bombus bimaculatus*, *B. borealis*, *B. citrinus*, *B. fervidus*, *B. griseocollis*, *Bombus impatiens*, *B. insularis*, *B. perplexus*, *B. rufocinctus*, *B. sandersoni*, *B. ternarius*, *B. terricola*, and *B. vagans* [50]. However, since some of these species can be difficult to identify in the field without a stereomicroscope, captured queens were grouped into morpho-species (grouping of morphologically similar species). The *B. borealis* group included *B. borealis* Kirby and *B. fervidus* (Fabricus), the *B. bimaculatus* group included *B. bimaculatus* Cresson and *B. griseocollis* (Degeer), and the *B. vagans* group included *B. vagans* Smith, *B. perplexus* Cresson, *B. sandersoni* Franklin, and *B. griseocollis* (this species can be placed in both groups). The other species encountered were more easily distinguishable and included *B. citrinus* (Smith), *B. impatiens* Cresson, *B. rufocinctus* Cresson, *B. ternarius* Say, and *B. terricola* Kirby. Identification was either performed upon marking or later based the color patterns noted in the field and with the help of a bumble bee identification guide [50]. 

### 2.3. Apple Quality

To quantify the effect of landscape enhancements on apple quality, 30 floral bouquets of Cortland apples—the only variety present in all the orchards—were marked with a flag during spring in 2018 and 2019. During the following fall season, the apple closest to each flag was selected, resulting in 30 apples collected in each of the 11 sites in 2018 and the 10 sites in 2019. The selected apples were collected from all orchards within two days and placed in a refrigerator at 4 °C overnight. Bags of apples were removed from the refrigerator sequentially, in order to prevent bias due to post-harvest maturation. Every apple was weighed (METTLER TOLEDO, Model ML 1502E, 0.01 g), the diameter (ROK digital caliper, Model 28123, 0.02 mm) and sugar level (ATAGO, Model 3810, 0–53%; Brix) were measured, and the number of seeds was counted. The number of seeds was used to evaluate apple shape [7,51]. 

### 2.4. Landscape Enhancements and Orchard Management

As few orchards in the study area had established landscape enhancements, several different types had to be included in our study (Table 1). Nevertheless, all enhancements could provide nesting, hibernating, and food resources for bumble bee queens, and no distinction was made among them for the analyses. All landscape enhancements were at least three years old and already well established. Flower strips on orchards #3 and 7 were mostly composed of native species of Asteraceae and Apiaceae. Orchard #4 had squashes intercropped with mature deciduous trees (maple: *Acer* spp. L., Sapindaceae; oak: *Quercus* spp. L., Fagaceae; and walnut: *Juglans* spp. L., Juglandaceae) aligned next to the apple trees. The windbreaks of orchards #2, 4, 7, and 8 were mostly composed of a mix of deciduous (maple, oak, walnut, and poplar: *Populus* spp. L., Salicaceae) and coniferous (pine: *Pinus* spp. L., Pinaceae; spruce: *Picea* spp., Pinaceae; and balsam fir: *Abies balsamea* (L.) Mill., Pinaceae) trees.

All orchards were managed conventionally, except orchards #2 and 6, which were organically managed. We estimated the potential toxic load that may be harmful to bumble bees based on the pesticide registries that orchards provided to the research team for 2017 (Appendix A). We first determined the toxicity to bees [52] of each pesticide used on a scale ranging from 1 (low toxicity) to 3 (highly toxic). We then calculated the intensity of pesticide use as the number of applications A of each pesticide *i* multiplied by its toxicity level T and summed that product over all n pesticides used on a given orchard:*Intensity of pesticide use* = ∑^n^_*i*=1_ (*T_i_* × *A_i_*)(1)

For the purposes of analyses, we converted the intensity data into a binary variable reflecting pesticide intensities ≤48.5 (low) and >48.5 (high) to provide a more balanced dataset.

### 2.5. Statistical Analyses

#### 2.5.1. Bumble Bee Community

The capture–mark–recapture (CMR) data we collected were originally intended to estimate bumble bee abundance while accounting for imperfect detection probability [53]. However, some species were captured in low numbers and the level of recapture of all species was too low (<5%) to perform CMR analyses based on individual captures. We thus converted the original CMR information into detection data whereby species *i* was either detected (1) or undetected (0) at site *j* during visit *k.* We then used a Bayesian hierarchical community model to estimate the impact of landscape enhancements on species-specific site occupancy, community composition, and species richness [54,55,56]. This model simultaneously estimates the effects of explanatory variables on species occupancy (φ) and detection probability (*p*) using logit link functions similar to logistic regression [57,58]. We allowed the occupancy of species *i* at site *j* to vary with season (spring vs. fall), management (with enhancements vs. none), and pesticide use intensity (low vs. high), as well as site:logit(ψ*_ij_*) = φ*_i_* + β_Season_ *_i_* × Season*_j_* + β_Management_ *_i_* × Enhancement*_j_* + β_Intensity_ *_i_* × Pesticide_j_ + α*_j_*(2)
where φ*_i_* corresponds to the random intercept of occupancy associated with species *i*, normally distributed with hyperparameters mean μ_φ_ and standard deviation σ_φ_ (φ*_i_* ~ N(μ_φ_, σ_φ_)). The slopes β*_i_* of season, enhancement, and pesticide use intensity were allowed to vary for each species *i*, normally distributed with mean μ_β_ and standard deviation σ_β_ (β*_i_* ~ N(μ_β_, σ_β_)). A random effect of site (α*_j_*) was included to account for multiple observations from the same site across different seasons, and this parameter was normally distributed with mean 0 and standard deviation σ_site_ (α*_j_~*N(0, σ_site_)).

We modeled the detection probability of species *i* at site *j* during visit *k* as a function of the air temperature and time of day during the visit:logit(*p_ijk_*) = η*_i_* + β_Air i_ × Air*_ijk_* + β_Time_ *_i_* × Time_ij*k*_(3)
where η*_i_* is a species random intercept associated with species *i* and follows a normal distribution η*_i_ ~* N(μ_η_, σ_η_). The slopes β*_i_* of air temperature and time of day were allowed to vary for each species *i*, normally distributed with mean *μ_β_* and standard deviation σ_β_
*(*β*_i_ ~* N(μ_β_, σ_β_)). The complete code for the model is provided in Appendix A.

Parameters of the community occupancy model were estimated by Markov chain Monte Carlo (MCMC) using five chains [59]. Each chain was run with 500,000 iterations, using 250,000 iterations as burn-in, and a thinning rate of 10. We used vague prior distributions for all parameters. Specifically, we used normal priors with N(0, 1000) for the β parameters and uniform priors U(0, 50) for all standard deviation parameters, except for the variance of site random effect where it was U(0, 150). This model was implemented in JAGS 4.3.0 within R 4.0.2 with the jagsUI and coda packages [60,61,62,63]. We used trace plots, posterior density plots, and the Brooks–Gelman–Rubin statistic to assess convergence. We assessed model fit using posterior predictive checks with a Pearson chi-square aggregated over rows and columns [55]. We also computed the area under the receiver operating characteristic (ROC) curve as a measure of predictive ability [64], where values of 0.5 indicate that the discriminatory ability of the model is not better than random and values > 0.5 indicate improvement in discrimination up to a maximum of 1 (perfect classification). We report means and 95% credible intervals (95% CRI) around the parameters, where intervals excluding 0 denote that the effect of a variable differs from 0.

#### 2.5.2. Apple Quality

The effect of landscape enhancements on apple characteristics was evaluated using generalized linear mixed models with a normal distribution and identity link for apple weight, diameter, and sugar level, and with a Poisson distribution and a log link, for the number of seeds in each apple [65]. Similar to the community occupancy model above, we included management and pesticide use intensity as explanatory variables. Season was not included in the analysis because apples were collected in the fall and preliminary analysis showed no difference between years. We included orchard identity as a random effect and estimated the model parameters with MCMC based on 5 chains, each consisting of 250,000 iterations with the first 125,000 as a burn-in period and a thinning rate of 10. We used N(0, 1000) priors for the β parameters and uniform priors U(0, 100) for all standard deviation parameters (Appendix A). We computed the Pearson chi-square as a measure of fit and used Pearson residuals to check for departures from model assumptions. As above, analyses were conducted in JAGS and with the same convergence diagnostics. 

## 3. Results

During the three years of the project, a total of 4765 individual queens were captured, with an average of 1453 captures per spring and 214 captures per fall across the 12 orchards. On average (±SD), 119 ± 106 and 17 ± 20 queens were caught per orchard in the spring and fall, respectively. The large seasonal difference likely resulted from the ease of finding bumble bees in the spring when apple trees were in bloom, as opposed to the fall, when floral resources were scarce. A total of eight species or morpho-species were captured. By far the most abundant species was *Bombus impatiens*, followed by the group *B. bimaculatus* and *B. ternarius* (Figure 3).

### 3.1. Bumble Bee Community

The hierarchical community model included the eight species detected during our study. Trace plots and the Brooks–Gelman–Rubin statistic suggested model convergence (R-hat < 1.06 for all parameters) and that chains were sufficiently long (MCMC error < 2% of the posterior standard deviation, sensu [61]). There were no indications that the model lacked fit, based on either the posterior predictive check on rows or on columns (χ^2^_rows_ = 37.05, *p* = 0.47; χ^2^_columns_ = 596.39, *p* = 0.39). The area under the ROC curve was 0.80 (95% CRI: [0.62, 0.89]), suggesting good model performance. 

Species richness was greater in spring than in fall (Number of Species_Spring_—Number of Species_Fall_ = 2.81, 95% CRI: [1.86, 3.81]). A greater number of species occurred at orchards with enhancements than at those without enhancements (Number of Species_Enhancement_—Number of Species_NoEnhancement_ = 1.10, 95% CRI: [0.59, 1.71]). Species richness was higher at orchards with low levels of pesticide use compared to orchards with higher levels of pesticide use (Number of Species_LowPesticide_—Number of Species_HighPesticide_ = 0.92, 95% CRI: [0.43, 1.50]).

Site occupancy by B. terricola, B. bimaculatus, and B. vagans was lower in the fall than in spring (Figure 4A). Bombus ternarius followed the same pattern, although the difference was marginal (90% CRI: [−2.36, −0.08]). Site occupancy by *B. impatiens*, *B. bimaculatus*, and *B. vagans* was higher in orchards with enhancements than those without (Figure 4B). *Bombus ternarius* was the only species whose site occupancy responded to the intensity of pesticide use, and it was lower in orchards with high levels of pesticide use than in orchards with lower levels of pesticide use (Figure 4C). We found no effect of the covariates we considered on the site occupancy of *B. borealis*, *B. rufocinctus*, and *B. citrinus*. The detection probability of all species increased with the number of hours after sunrise (Figure 5A). However, the detection probability did not vary with air temperature, regardless of species (Figure 5B).

### 3.2. Apple Quality

Markov chain Monte Carlo diagnostics indicated that the generalized linear mixed models on apple characteristics converged (trace plots, R-hat < 1.0002) and that chains were sufficiently long (MCMC error < 0.5% of the posterior standard deviation). Residual diagnostics and posterior predictive checks suggested model fit (*p* value: 0.50–0.96). Apple weight was higher in orchards with high pesticide use than in orchards with lower levels of pesticide use (β_HighIntensity_ = 29.24, 95% CRI: [0.22, 61.90]) but did not vary with the presence of landscape enhancements (β_LandscapeEnhancement_ = 7.81, 95% CRI: [−20.92, 44.25]). In contrast, apple diameter did not vary either with landscape enhancements (β_LandscapeEnhancement_ = −0.68, 95% CRI: [−6.31, 5.32]) or with the intensity of pesticide use (β_HighIntensity_ = 3.98, 95% CRI: [−1.65, 9.97]). The sugar level in apples followed the same pattern (β_LandscapeEnhancement_ = −0.04, 95% CRI: [−1.05, 0.96]; β_HighIntensity_ = 0.60, 95% CRI: [−0.40, 1.61]). Finally, the number of seeds was lower at sites with landscape enhancements than at sites without enhancements (β_LandscapeEnhancement_ = −0.20, 95% CRI: [−0.37, −0.02]) but did not vary with the intensity of pesticide use (β_HighIntensity_ = 0.03, 95% CRI: [−0.15, 0.20]). 

## 4. Discussion

### 4.1. Landscape Enhancements

In support of our hypothesis, we found that the site occupancy of three species (groups) of bumble bee queens, namely, *B. impatiens*, group *B. bimaculatus*, and group *B. vagans*, was higher in orchards with landscape enhancements than in orchards without such structures (Figure 4B). These results are significant as they are, to our knowledge, the first that account for the imperfect detection of bees when evaluating the effects of landscape enhancements on these insects. Moreover, they are consistent with several reports of bumble bee abundance being higher in conventionally managed winter wheat fields with adjacent flower strips, compared to conventional fields and organic fields without flower strips [37]. There is evidence that flower strips have multiple positive effects on pollinators in agroecosystems, even more so when those strips are diverse [36,37,66]. For instance, the presence of hedgerows in raspberry, blueberry, and apple orchards in southern Quebec was beneficial to wild bee abundance, in terms of the availability of both floral and nesting resources [41]. Hedgerows were also shown to act as corridors for bumble bee dispersal [67] and to contribute to a more homogenous distribution of native pollinators in blueberry fields [68]. However, contrary to flower strips, which represent a generally positive presence for bees [69], hedgerows may be beneficial only under certain contexts. For instance, hedgerows were visited by foraging and nest-searching bumble bee queens less often than arable field margins in Scotland [43]. Further research regarding the benefits of the different types of landscape enhancements and how these benefits may vary according to the surrounding landscape structure is thus warranted.

The increased presence of bumble bees in orchards that we documented can potentially be financially rewarding for apple growers. Indeed, Quebec apple producers generally pay large sums of money to rent hives of honey bees, a species not native this region, to ensure the pollination of apple trees [10]. Orchards with a high abundance of bumble bees could probably reduce or even eliminate the need to rent honey bee hives as bumble bees are much more efficient pollinators and work longer and at cooler temperatures than honey bees [11]. 

As expected, landscape enhancements had a positive effect on bumble bee queen richness. This result is consistent with that of other studies showing a positive effect of either flower strips or hedgerows on bee species richness [37,40,68]. For example, an average of ~4.5 native bee species were found in conventional orchards with flower strips, in comparison to ~0.7 and ~2 species in conventional and organic orchards without flower strips, respectively [37]. Similar positive results were found for hedgerows, whose presence contributed to a more homogenous distribution of native bee species across blueberry fields [68]. Native bee species richness in hedgerows was also equal to or greater than species richness in adjacent hay crops and woodlots [40]. The greater bumble bee diversity found in orchards with landscape enhancements can be beneficial for growers since species diversity is usually associated with an increased resilience of ecosystems [70,71,72,73]. Such increased resilience could be particularly useful for ensuring pollination following major environmental disruptions, such as the ones expected with climate changes.

Contrary to our hypothesis, we found no effect of landscape enhancements on apple characteristics, except for the number of apple seeds which was lower at sites with landscape enhancements. However, these results are not unlike those found in the literature, despite not explicitly investigating apple yield. Flower strips generally increased the abundance and richness of pollinators, but this increase did not consistently lead to increased yield [74]. In fact, a meta-analysis showed that flower strips enhanced pest control by 16% in adjacent fields, but the effect on pollination was more variable and depended on landscape context [74]. Furthermore, flower diversity and older flower strips usually increased pollination services in apple orchards, but apple yield did not follow the same pattern [74]. Similar results were found on cucumber farms, where flower strips enhanced the presence of beneficial insects within fields without affecting total yield [75]. In our study, the 30 apples collected by orchard may have been insufficient to evaluate apple quality for the whole orchard, and thus the difference between orchards with and without landscape enhancements may not have been adequately measured. Furthermore, the use of only one variety (Cortland) might have limited our power to detect an effect at the orchard’s level. Unfortunately, more apples or more varieties could not be processed due to logistical limitations. Another study covering more sites and collecting more apple/variety samples per site might yield different results. 

### 4.2. Intensity of Pesticide Use

Despite the growing body of evidence documenting negative impacts of pesticides on bumble bees [26,76,77,78,79], and contrary to one of our hypotheses, we found a negative effect of intensity of pesticide use on the site occupancy by bumble bee queens for only one of the eight (groups of) species we detected in orchards, namely, *B. ternarius* (Figure 4C). Aside from the fact that *B. ternarius* was the third most captured species in our study (Figure 3) and that this may have led to an improved capacity of detecting such an effect, it remains that this result may also stem from interspecific differences in exposure or sensitivity to pesticides. Indeed, some species are more sensitive to certain threats due to their behavioral or functional traits, particularly when their functional traits are associated with narrow climatic specialization, late-emerging queens, and when the site is located closer to the species’ climatic tolerance range [80,81,82]. For instance, *B. ternarius* is known to nest underground [50], and this functional trait may thereby increase its exposure to pesticides in orchards since pesticides can accumulate and reach lethal concentrations for bees in the soil [83]. This being said, the fact that one of the most abundant species in the study area, *B. ternarius*, is significantly impacted by pesticides should raise concern. Known threats that are not mitigated or eliminated rapidly enough can indeed seriously impair the population persistence of bumble bee species [84,85]. One such example involves the loss of habitat, a well-known cause of species decline, that drove *B. affinis* to near extinction in eastern North America [84,85]. 

As expected, species richness was negatively influenced by the intensity of pesticide use. As mentioned above and shown here, evidence of negative effects of pesticides on bumble bee species, such as *B. impatiens* and *B. terrestris,* is mounting [26,76,77,78,79]. However, studies looking at the effects of pesticides on bee communities remain scarce, and those that did so while controlling for imperfect species detection are even scarcer. Nonetheless, one of these previous studies found results similar to ours, yet for the whole wild bee community: fewer species were observed at the field, landscape, and regional (river basins) scales [86]. At the field scale, wild bee communities were less species-diverse when pesticides were applied more than twice during the summer [86]. At the landscape level, pesticide-treated vine fields were less species-diverse than uncultivated fields and maize fields [86]. Lastly, bumble bee species were less species-diverse in the more intensively farmed basins [86]. Future research should definitely assess whether landscape enhancements can mitigate the impact of pesticide use on pollinator communities and provide a source of predators and parasitoids to control pest insects. 

Finally, apart from a positive effect on apple weight, we found no effect of the intensity of pesticide use on the three other characteristics related to apple quality that we considered (diameter, sugar level, and number of seeds). This is surprising given that pesticides are used to increase yield by, among other things, enhancing individual apple quality. Although our sampling effort may have been too small (e.g., 30 apples of one variety per orchard) to detect an effect, other studies in cucumber and watermelon cropping systems also failed to find a relationship between crop yield and pesticide use [87,88]. In fact, even though cucumber yields were greatly improved when pollination was increased, they were not enhanced when pesticides were used for pest control [88]. Similarly, while increased use of fertilizers and irrigation did not increase watermelon yields, adequate pollination did [87]. Furthermore, we found a relationship with apple weight, but not apple diameter. This could suggest that apples produced under more intensive management might be denser. However, we did not investigate that surprising result further as it was outside of the scope of this project. More research on the cascading effects of pesticides on pollinator communities and, in turn, of the latter on pollination and crop quality are needed.

## 5. Conclusions

This study aimed at determining if landscape enhancements could help apple growers improve bumble bee queen presence as well as apple quality in their orchards. The impact of intensity of pesticide use on both bumble bee queens and apple quality was also evaluated. Landscape enhancements had a positive effect on bumble bee queen site occupancy and species richness, but they had no effect on four characteristics representing apple quality. While the intensity of pesticide use was negatively associated with *B. ternarius* site occupancy, it was only positively associated with apple weight. To our knowledge, this study is one of the first to assess the influence of landscape-related effects on bumble bee site occupancy and species richness in agroecosystems while controlling for imperfect species detection (see also [56,89]). Our work thus provides key additional evidence that landscape enhancements improve bumble bee queen presence and diversity in apple orchards and should therefore be considered by growers as a means to enhance and ensure the pollination and diversity of beneficial insects in their orchards.

## Figures and Tables

**Figure 1 insects-12-00421-f001:**
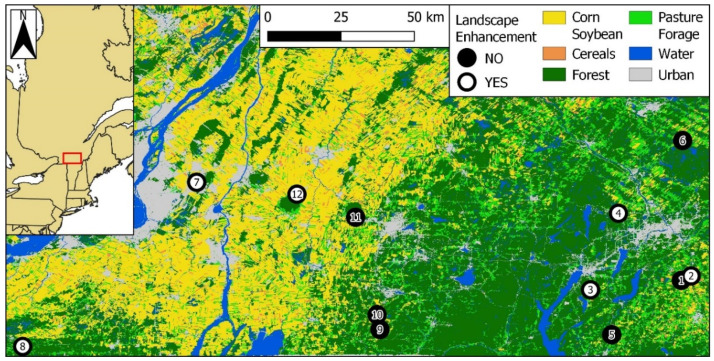
Apple orchards (n = 12) wherein bumble bee communities were monitored between 2017 and 2019 in southern Quebec, Canada. Orchards in the western cluster are in the Montérégie-Est region (#7 to 12); those in the eastern cluster are in the Estrie region (#1 to 6). White and black dots represent orchards with and without landscape enhancements, respectively. Orchard ID numbers are included in the dots. Coordinate system: WGS84 (EPSG 4326). Source: underlying raster based on the Annual Crop Inventory of Agriculture and Agri-food Canada [45].

**Figure 2 insects-12-00421-f002:**
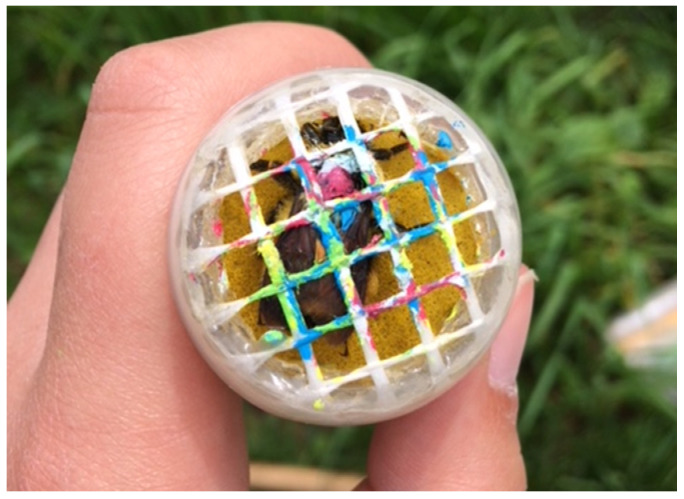
Bumble bee queen in a marking cage after being assigned a 3-color combination (i.e., white, red, and blue). Picture taken by A. Gervais.

**Figure 3 insects-12-00421-f003:**
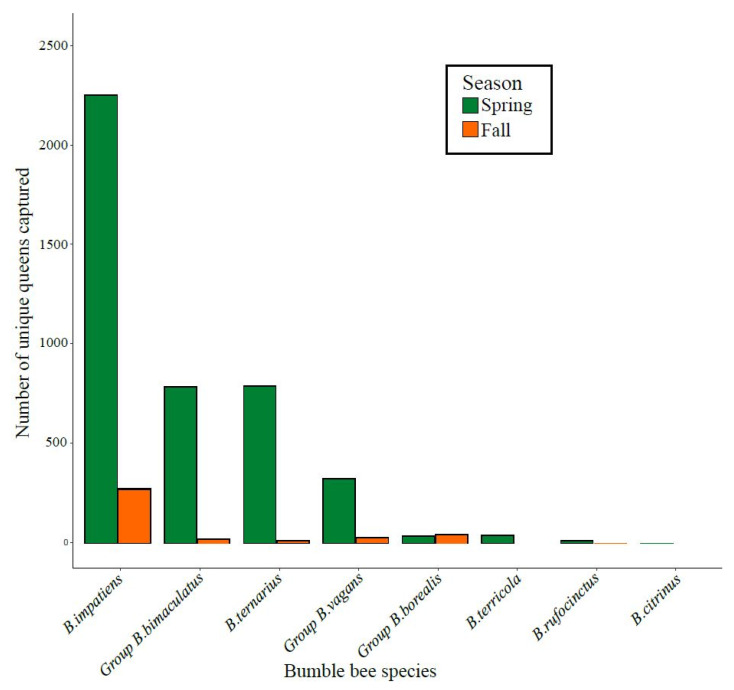
Total number of individual queens of the different species or morpho-species caught in apple orchards in fall or spring across the three years of the project (2017–2019).

**Figure 4 insects-12-00421-f004:**
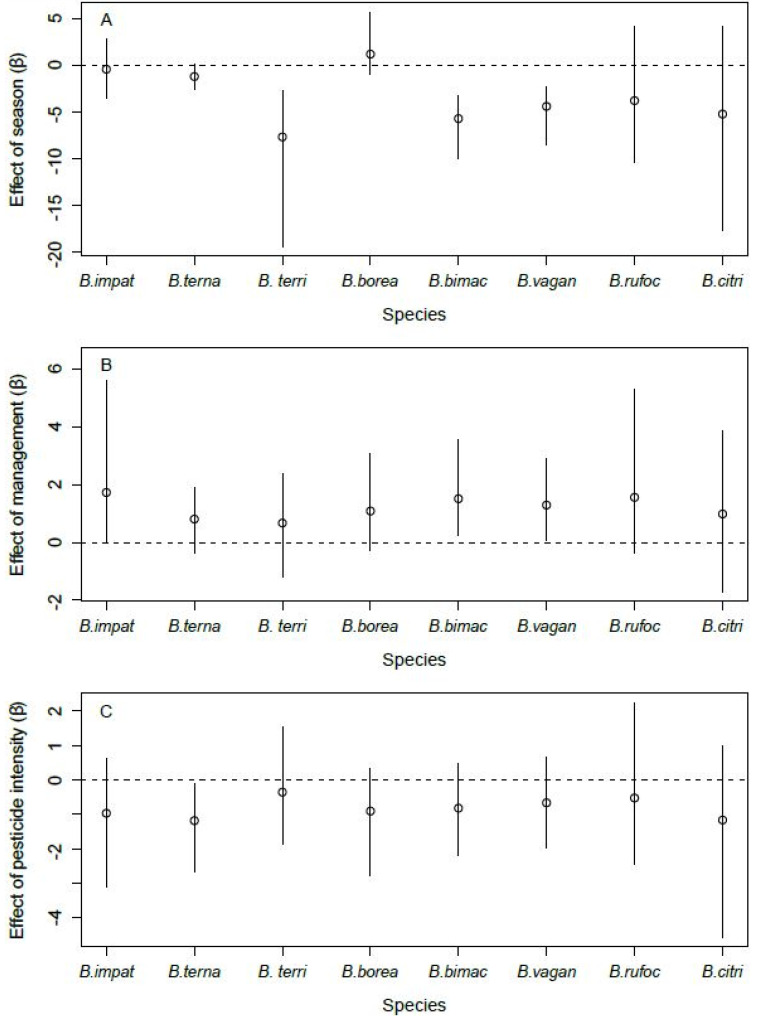
Effect of season ((**A**), spring vs. fall), management ((**B**), with enhancements vs. none), and intensity of pesticide use ((**C**), low vs. high) on the site occupancy of the eight species of bumble bee queens sampled in orchards (2017–2019) in southern Québec, Canada. Error bars denote 95% Bayesian credible intervals around each estimate.

**Figure 5 insects-12-00421-f005:**
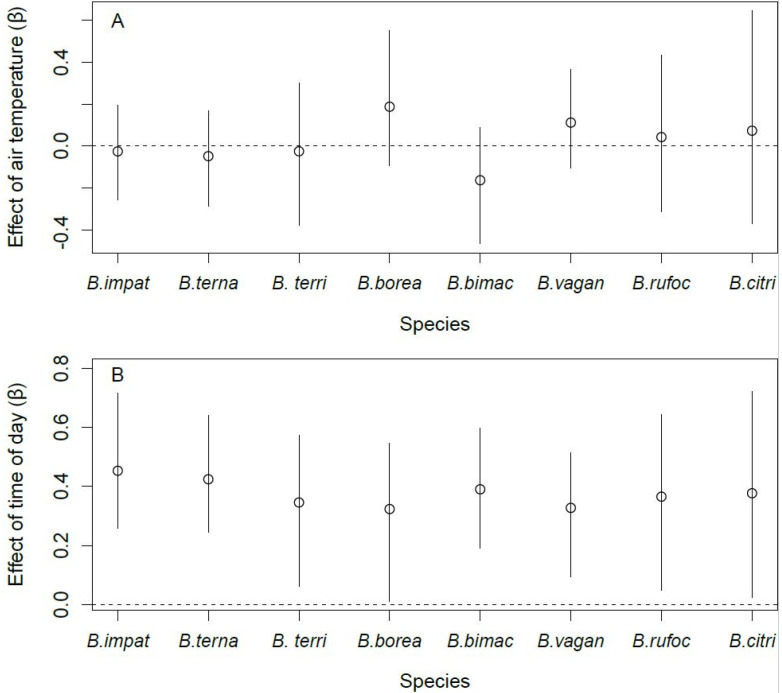
Effect of air temperature (**A**) and time of day (**B**) on the detection probability of the eight species of bumble bee queens sampled in orchards (2017–2019) in southern Québec, Canada. Error bars denote 95% Bayesian credible intervals around each estimate.

**Table 1 insects-12-00421-t001:** Characteristics of the 12 orchards sampled from 2017 to 2019. The orchard numbers correspond to those in Figure 1. Intensity of pesticide use was defined according to Equation (1) (see text).

# Orchard	Region	Type of Landscape Enhancements	Index of Pesticide Use (0 = Low; 109 = High)
1	Estrie	None	23
2	Estrie	Deciduous/coniferous windbreaks	13
3	Estrie	Flower strips	27
4	Estrie	Deciduous/coniferous windbreaks and intercropping	51
5	Estrie	None	48
6	Estrie	None	49
7	Montérégie	Deciduous/coniferous windbreaks and flower strips	109
8	Montérégie	Deciduous/coniferous windbreaks	32
9	Montérégie	None	70
10	Montérégie	None	54
11	Montérégie	None	52
12	Montérégie	Coniferous windbreaks	48

## Data Availability

The data presented in this study will be openly available in Scholars Portals Dataverse with a valid DOI number once published.

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
