# Peer review of "Landscape Enhancements in Apple Orchards: Higher Bumble Bee Queen Species Richness, but No Effect on Apple Quality"

_insects, 2021, doi:10.3390/insects12050421_

Round 1
Reviewer 1 Report
The study Landscape enhancements in apple orchards: higher bumble bee 2 queen species richness, but no effect on apple quality by Grevais et al., is focused on bumble bee diversity and apple quality across 12 apple orchards located in Quebec, Canada with observations spanning a two year period, in the fall and winter. The subject of this study is of interest due to the importance of biodiversity, pollinator health and sustainable agriculture. This study supports the idea that increased landscape floral diversity surrounding agricultural areas can increase bumble bee species diversity and quantity of individuals. It would add to the scientific merit of this study if the authors would have included additional data in their analysis such as what pesticides were used in the orchards and application times.
It has been shown in multiple studies pesticide class and even different compounds in the same pesticide class have varying degrees of toxicity towards bees in addition to what combinations of different classes are used, even including time of application would benefit the analysis. Including which pesticides were used during the course of these type of studies could benefit future pesticide usage practices. Additionally, if the same pesticides were used in the none landscape managed orchards and the landscape managed ones, the authors argument that the resulting study outcome supports their hypothesis would be stronger. If any of this information was collected it would make a nice addition to this interesting study but if not, it is highly recommended that authors attempt to include it in future studies.
Some minor suggestions:
- Line 69. Are you saying bumble bee belongs to genus Andrena?
- Figure 1: 1 to 6 are on the east side of the map, says western in legend
- Line 180: you say 2008, typo?
- Figure 3: Resolution of this figure is bad. Hard to make out axis labels. Also, it would be a more clear representation of data, if bars for fall and spring were placed side by side and not stacked on top of each other. As presented it could be read as if fall is greater than spring, but that is not what you say in the text.
- Figures 4 and 5 need increased resolution, hard to read as is.
- Line 414: limited
- Line 453: Could help the reader if the additional three apple qualities are relisted in parenthesis. Also, in one part of the manuscript the authors say weight and in others say mass. Technically this two things are different. An additional question about apple quality, a positive effect on mass was observed but diameter was unchanged? So are those apples more dense? You state no effect on sugar. What is causing the increase in mass measurement? Water? Tissue density? Also, you discuss yield in this paragraph. You state you only checked 30 apples per orchard. I think in regards to yield, to make any conclusion for this particular study in regards to pesticide usage and yield, you need some data on actual crop yield.
Author Response
Dear Reviewer 1,
We would like to thank you for your comments. Below you will find our answers to your comments.
Cordially,
Amélie
Comment: The study Landscape enhancements in apple orchards: higher bumble bee 2 queen species richness, but no effect on apple quality by Grevais et al., is focused on bumble bee diversity and apple quality across 12 apple orchards located in Quebec, Canada with observations spanning a two year period, in the fall and winter. The subject of this study is of interest due to the importance of biodiversity, pollinator health and sustainable agriculture. This study supports the idea that increased landscape floral diversity surrounding agricultural areas can increase bumble bee species diversity and quantity of individuals.
Response: Thank you for these comments.
Comment: It would add to the scientific merit of this study if the authors would have included additional data in their analysis such as what pesticides were used in the orchards and application times. It has been shown in multiple studies pesticide class and even different compounds in the same pesticide class have varying degrees of toxicity towards bees in addition to what combinations of different classes are used, even including time of application would benefit the analysis. Including which pesticides were used during the course of these type of studies could benefit future pesticide usage practices. Additionally, if the same pesticides were used in the none landscape managed orchards and the landscape managed ones, the authors argument that the resulting study outcome supports their hypothesis would be stronger. If any of this information was collected it would make a nice addition to this interesting study but if not, it is highly recommended that authors attempt to include it in future studies.
Response: We fully agree with the reviewer. Yet, only the list of pesticides used in the different orchards were made available to us as this is sensitive information. We now provide the list of pesticides used by the different “types” of orchards in a supplementary table (S1).
Comment: Line 69. Are you saying bumble bee belongs to genus Andrena?
Response: We meant that Bombus AND Andrena are both really efficient genus for apple pollination. However, we removed it as it was confusing and modified the sentence accordingly. See line 69.
Comment: Figure 1: 1 to 6 are on the east side of the map, says western in legend
Response: Thank you for catching this error. We corrected the legend accordingly. It now reads: “ Orchards in the western cluster are in the Montérégie-Est region (#7 to 12), those in the eastern cluster are in the Estrie region (#1 to 6) “. We also modified Figure 1 in order to provide more and clearer information regarding the landscape context, notably with respect to agriculture and forest cover, in which the different orchards were located. We hope you will approve of this.
Comment: Line 180: you say 2008, typo?
Response: Yes, it was supposed to be 2018. Correction made. Thank you.
Comment: Figure 3: Resolution of this figure is bad. Hard to make out axis labels. Also, it would be a more clear representation of data, if bars for fall and spring were placed side by side and not stacked on top of each other. As presented it could be read as if fall is greater than spring, but that is not what you say in the text.
Response: We increased the label size and the resolution. We also placed, as suggested, the bars side by side.
Comment: Figures 4 and 5 need increased resolution, hard to read as is.
Response: We uploaded figures in .pdf format with increased resolution.
Comment: Line 414: limited
Response: Corrected, thank you. Line: 420
Comment: Line 453: Could help the reader if the additional three apple qualities are relisted in parenthesis.
Response: There were four apple quality metrics: apple weight, diameter, sugar level, and number of seeds. We added these four apple qualities in the sentence on lines 458-460: “ Finally, apart from a positive effect on apple weight, we found no effect of the intensity of pesticide use on the three other characteristics related to apple quality that we considered (diameter, sugar level, and number of seeds).” We also modified the results section to clarify this confusion on Lines 360-363: “Finally, the number of seeds was lower at sites with landscape enhancements than without enhancements (βLandscapeEnhancement = -0.20, 95% CRI: [-0.37, -0.02]), but did not vary with the intensity of pesticide use (βHighIntensity = 0.03, 95% CRI: [-0.15, 0.20]).”
Comment: Also, in one part of the manuscript the authors say weight and in others say mass. Technically this two things are different.
Response: We removed mass and replaced it with weight throughout the manuscript.
Comment: An additional question about apple quality, a positive effect on mass was observed but diameter was unchanged? So are those apples more dense? You state no effect on sugar. What is causing the increase in mass measurement? Water? Tissue density?
Response: That is an excellent comment. We searched the literature but could not find an explanation. We added the following sentence on line 468-471 to highligth the result: Furthermore, we found a relationship with apple weight, but not apple diameter. This could suggest that apples produced under more intensive management might be dens-er. However, we did not investigate that surprising result farther as it was outside of the scope of this project
Comment: Also, you discuss yield in this paragraph. You state you only checked 30 apples per orchard. I think in regards to yield, to make any conclusion for this particular study in regards to pesticide usage and yield, you need some data on actual crop yield.
Response: We agree with the reviewer. We clarified this situation in the text on line 408-409: However, these results are not unlike those found in the literature, despite not explicit-ly investigating apple yield”.
Reviewer 2 Report
The research design was of high quality. The statistical analysis was excellent.
It is suggested that the comment about the mass of the apples being greater in the orchards where pesticide use was greatest be deleted from the Abstract. Because it is likely this result is due to random error (not a real effect and not due to any errors in study design), mentioning this in the Abstract draws attention to a spurious result that will not be seen in future research.
Author Response
Dear Reviewer 2,
We would like to thank you for your comments. Below you will find our answers to your comments.
Cordially,
Amélie
Comment: The research design was of high quality. The statistical analysis was excellent. It is suggested that the comment about the mass of the apples being greater in the orchards where pesticide use was greatest be deleted from the Abstract. Because it is likely this result is due to random error (not a real effect and not due to any errors in study design), mentioning this in the Abstract draws attention to a spurious result that will not be seen in future research.
Response: We would like to thank the reviewer for its kind comments. We however disagree with the reviewer concerning deleting the sentence from the abstract. We think the apple quality part should still be included in the abstract, since we analyzed the apples and discussed our results. Yes, we only used 30 apples per orchards, but we mentioned this weakness in our manuscript (LINES:416-419) and the results can still be of interests for someone studying apple qualities.
Reviewer 3 Report
The authors present a vert well-done study on the effect of apple orchard enhancement on richeness and occupancy of bumbleebees in Canada. I have really enjoyed this paper. The topic is of absolute interest, the presentation is very clear, the Results are important and well discussed. Maybe the only minor problem is that identification of bumblebee species could not be assessed in detail. In this sense it would be nice if the authors have previous data on the species occurring in their area and cite this in the ms. Apart from this, I have no particular comments, really it is one of the best-done manuscripts I have reviewed in the last months. There are some minor spelling problems (some italics for species are lacking, for example), but really very few things.
Author Response
Dear Reviewer 3,
We would like to thank you for your comments. Below you will find our answers,
Cordially,
Amélie
Comment: The authors present a very well-done study on the effect of apple orchard enhancement on richness and occupancy of bumblebees in Canada. I have really enjoyed this paper. The topic is of absolute interest, the presentation is very clear, the Results are important and well discussed. Maybe the only minor problem is that identification of bumblebee species could not be assessed in detail. In this sense it would be nice if the authors have previous data on the species occurring in their area and cite this in the ms. Apart from this, I have no particular comments, really it is one of the best-done manuscripts I have reviewed in the last months. There are some minor spelling problems (some italics for species are lacking, for example), but really very few things.
Response: We thank the reviewer for the kind comments. We double-checked all the species names to make sure they were in italics. We also added the list of the species occurring in our study area according to the work of Williams et al. (2014) and the extensive sampling conducted on 40 farms of our study area between 2006 and 2016 (Gervais, 2020). Lines 166-168
Gervais, 2020. De la communauté à l’individu: influence de l’intensité agricole et du paysage sur les bourdons (Bombus spp.) du Sud du Québec. PhD Thesis. https://corpus.ulaval.ca/jspui/bitstream/20.500.11794/66288/1/35718.pdf